# Tripeptide Leu-Ser-Trp Regulates the Vascular Endothelial Cells Phenotype Switching by Mediating the Vascular Smooth Muscle Cells-Derived Small Extracellular Vesicles Packaging of miR-145

**DOI:** 10.3390/molecules27207025

**Published:** 2022-10-18

**Authors:** Tianyuan Song, Minzhi Zhou, Wen Li, Lin Zheng, Jianping Wu, Mouming Zhao

**Affiliations:** 1School of Food Science and Engineering, South China University of Technology, Guangzhou 510640, China; 2Chaozhou Branch of Chemistry and Chemical Engineering Guangdong Laboratory, Chaozhou 521000, China; 3Department of Agricultural, Food and Nutritional Science, University of Alberta, Edmonton, AB T6G 2P5, Canada

**Keywords:** soybean protein-derived peptides, extracellular vesicles, miRNAs, HUVECs, atherosclerosis

## Abstract

Tripeptide LSW, initially identified as a potent ACE inhibitory peptide from soybean protein, was recently reported to exert a protective effect against angiotensin II-induced endothelial dysfunction via extracellular vesicles (EVs). However, the molecular mechanisms, especially in lipid accumulation-induced atherosclerosis, still remain unclear. The study aimed to investigate whether the protective effects of LSW against endothelial dysfunction on vascular endothelial cells (VECs) was via vascular smooth muscle cells (VSMCs)-derived miRNA-145 packaged in EVs. The miRNA-145 was concentrated in EVs from LSW-treated VSMCs (LEVs), internalized into the HVUECs, and targeted the programmed cell death protein 4 (PDCD4) expression of HUVECs. Oxidized low-density lipoprotein (oxLDL) was applied to induce endothelial dysfunction in HUVECs; oxLDL-induced endothelial dysfunction in HUVECs was attenuated by PDCD4 knockout or LEVs incubation. The results of this study suggested a novel function of LSW as a regulator on the functional EVs from vascular cells in the oxLDL-induced atherosclerotic model.

## 1. Introduction

Atherosclerosis (AS) is a vital cause of the cardiovascular disease (CAD) and a serious threat to systemic cardiovascular homeostasis [1,2]. As a vascular disease, its occurrence and development are closely related to vascular aging, lipid accumulation-induced thrombosis, vascular inflammation, and ischemic stroke [3,4]. The vascular endothelial cells (VECs) and smooth muscle cells (VSMCs) could respond to stimulation from the extracellular matrix, the main component of vessels, which was proposed as a crucial mechanism of endothelial dysfunction, sub-endothelial mononuclear cells infiltration, aorta remodeling, lipid deposition, and fibrosis [5]. The accumulation and oxidative modification of low-density lipoprotein (oxLDL) provokes the VECs and induces atherogenesis by combining the specific receptor lectin-type oxidized LDL receptor 1 (LOX-1) [6]. The oxLDL can break through the endothelial barrier and aggregates in the sub-endothelium, where it promotes the differentiation of monocytes into macrophages, which attempt to phagocytose the oxLDL and become lipid-laden foam cells [7]. The lipid-laden foam cells were formed by macrophages endocytosing the oxLDL, which is regarded as an ominous event in the AS [8]. Hence, the oxLDL was used as an induction cytokine for the endothelial dysfunction of AS pathogenesis in the present study.

Vascular cells are mediated by various external environmental stimuli (such as fluid shear, chemical reagents, and cytokines) and cause relevant physiological phenotypic switching [9]. In recent years, the nanoscale membrane-bound vesicles named “extracellular vesicles” (EVs) have been identified as an important regulator of intercellular communication, due to their ability to transfer abundant cell-specific cargo that contains proteins, lipids, and nucleic acids, and release them into the extracellular microenvironment [10]. Of particular interest is the single-stranded and non-coding small RNAs (miRNAs) that can be transferred by EVs and regulate specific gene functions in recipient cells [11]. Numerous studies demonstrated that miRNAs are involved in almost all stages of vascular dysfunction in AS. Therefore, miRNAs are commonly served as a classical biomarker to monitor the physiological state of vascular cells [12]. A protein array analysis of small EVs reported that dendritic cells (DC)-derived small EVs could stimulate human umbilical vein endothelial cells (HUVECs) by activating the NF-κB signaling pathway, and tumor necrosis factor-α (TNF-α) on the membrane of small EVs should be mostly responsible for this process [9]. A clinic study found that miRNA-155 was overexpressed in the urinary-EVs from CAD patients and suppressed the anti-inflammatory signaling pathway in macrophages [13]. The VECs-derived EVs enriched with miRNA-126 promoted monocyte adhesion by targeting the vascular cell adhesion molecule 1 (VCAM-1) expression [14]. The miRNA-143 and miRNA-145 have been identified as the key functional molecules in the VSMCs-derived EVs [15]. A research reported that knockout of miRNA-143 and miRNA-145 would disturb vascular homeostasis by inducing VSMCs dysfunction in mice [16]. MiRNA-143 and miRNA-145 were overexpressed in the EVs from KLF2-transduced- and shear-stress-induced HUVECs, and those EVs targeted gene expression of VSMCs after internalizing into VSMCs [17]. However, the mechanism of the association between miRNA-145 and oxLDL-induced endothelial dysfunction is still unclear.

Growing evidence indicated that the biopeptides encrypted in natural protein sequences are able to regulate various physiological functions. In recent decades, a variety of bioactive peptides have been identified from different protein sources. Due to their significant health-beneficial activities in many chronic diseases including hypertension, hyperlipidemia, and diabetes, these peptides were considered as the leading compounds in development of nutraceuticals or functional foods [18,19,20,21]. The milk-derived peptides VPP and IPP were found to inhibit the TNF-α-induced endothelial inflammatory by down-regulating pivotal protein expression in the NF-κB signaling pathway [18]. The egg-derived peptides IRW and IQW exhibited antihypertensive effects by promoting vasorelaxant [19]. Peptides HGSEPFGPR, RPRYPWRYT, and RDGPFPWPWYSH isolated from amaranth (*Amaranthus hypochondriacus*) proteins were reported to reduce the low-density lipoprotein receptor-1 (LOX-1), intercellular cell adhesion molecule-1 (ICAM-1), and matrix metalloproteinase-9 (MMP-9) expression in the THP-1 macrophage, which implied that those peptides have potential to improve AS [20].

Our lab has been dedicated to understanding the relationship between the biopeptides and EVs-regulation, and our recent findings suggested that the soybean protein-derived tripeptide LSW was able to reverse the adverse effects of Ang II-induced VSMCs-secreted EVs on endothelial cells [21]. The transcriptomic analysis demonstrated that the EVs produced by Ang II-induced VSMCs aggravated the endothelial dysfunction through TNF, NF-κB, and NOD-like receptor signaling pathways. However, the mechanisms of miRNAs loading in EVs from LSW-induced VSMCs on endothelial cells were still unclear. Therefore, this study was aimed to investigate the effect of LSW on the expression of miRNA-145 in the VSMCs-derived EVs, and the oxLDL-induced HUVECs were used as an atherosclerosis model to explore the mechanism of tripeptides LSW.

## 2. Results

### 2.1. The Effects of Tripeptide LSW on VSMCs and HUVECs

The proliferation and migration of vascular cells are closely responsible for many vascular diseases such as atherosclerosis. Hence, the cck-8 and wound healing assay were performed to test the effects of LSW on the proliferation and migration of VSMCs and HUVECs. Figure 1 showed that the LSW has no effect on proliferation and migration at a gradient concentration from 0.1 to 100 μmol/L. Oxidative stress is the primary adverse reaction of vascular cells responding to external stimuli, which causes a large amount of ROS production and further damages cellular function. Dihydroethidium could be internalized into VSMCs and HUVECs, and intracellular superoxide anions dehydrogenates dihydroethidium to produce ethidium. The ethidium could bind to RNA or DNA to produce red fluorescence. Figure 1B showed no significant change of red fluorescence in the DHE-labeled VSMCs or HUVECs as the concentration of LSW gradually increased. Therefore, the LSW incubation has no effect on the amount of ROS produced within VSMCs and HUVECs. The DHE assay exhibited that LSW was a mild response to oxidative stress in cells (Figure 1B,C). The wound healing assay indicated that the LSW has no specific influence on the migration of HUVECs (Figure 1D,E). Previous studies considered that the LSW as an ACE inhibitor could inhibit the proliferation and migration of Ang II-induced VSMCs, and the data from our lab showed that LSW incubation could revise the anomalous loading of EVs in the Ang II-induced VSMCs [21,22]. Therefore, this result demonstrated that the LSW should be a safe response factor for VSMCs and HUVECs.

### 2.2. Characterization of EVs from VSMCs with or without LSW-Incubation

The EVs isolated from the medium of VSMCs with or without LSW-incubation were morphologically confirmed using TEM (Figure 2A). The EVs produced by normal VSMCs without LSW incubation were abbreviated as “NEVs”, and the LSW-induced VSMCs-derived EVs were abbreviated as “LEVs”. The EVs extracted through ultracentrifugation displayed a classical “cup-shaped” and double-layer membrane structure under an electron microscope, and the particle diameter was approximately 200 nm (Figure 2A). The results of NTA suggested the diameter and size distribution of two EVs (Figure 2B). In addition, CD9, CD81, and Tsg101 as protein markers of EVs were obviously detected by western blot, which further confirmed the identity of both NEVs and LEVs (Figure 2C). These results indicated that the LSW incubation has not intervened in the process of EV secretion. Remarkably, although the above features conformed to the specifications of “exosomes”, the vesicle sample in the present study was uniformly described as the term “EV” due to the indistinct specific markers of subcellular origin data according to MISEV2018 [23].

### 2.3. LEVs Attenuated the oxLDL-Induced Endothelial Dysfunction in HUVECs

The effects of NEVs and LEVs on endothelial dysfunction were explored. The model of endothelial dysfunction was prepared by oxLDL induction, and oxLDL significantly promoted the proliferation of HUVECs based on a cck-8 assay (Figure 3A). The wound healing experiment showed that the speed of scratch healing of oxLDL-induced HUVECs was conspicuously stronger than normal HUVECs, which suggested that oxLDL accelerated the migration capacity. However, LEVs but not NEVs treatment could observably attenuate the oxLDL-induced proliferation and migration of HUVECs (Figure 3B,C). It signifies that the NEVs and LEVs were similar in morphological characteristics but different in physiological functions. In addition, the pure peptide LSW has also performed the same pre-incubation operation, but the single incubation with LSW has no effect on the excessive proliferation and migration of oxLDL-induced HUVECs (Figure 3B,C). The monocytes adhesion assay showed that exposure to oxLDL increased the number of monocytes adhered to endothelial cells, which was reduced by LEVs incubation. Both NEVs and pure peptide LSW treatment had no effect on oxLDL-induced HUVECs (Figure 3D,E). The important adhesion molecules were further investigated, and the results indicated that LEVs could significantly reduce the expression of adhesion molecules VCAM-1, ICAM-1, and E-selectin in the oxLDL-stimulated HUVECs (Figure 3F). The results in Figure 3 demonstrated that the LEVs improved the endothelial dysfunction caused by oxLDL as pro-proliferation, pro-migration, and adhesion molecules up-regulation contrast to NEVs, which encouraged exploring the different molecules packaging between LEVs and NEVs.

### 2.4. LSW Promoted miR-145 Loading in VSMCs-Derived EVs and Targeted PDCD4

Numerous studies reported that miR-145 is responsible for phenotype modulation of VSMCs and can be transferred by EVs [15]. Hence, the miR-145 loading in EVs was quantified by qRT-PCR. As is shown in Figure 4, LSW incubation increased the expression of miR-145 in both VSMCs and VSMCs-derived EVs, but the different degree of miR-145 expression in EVs was much greater than in cells (Figure 4A). This result implied that the LSW could enhance the load of miR-145 into VSMCs-derived EVs. According to the TargetScan database (version: 7.2, http://www.targetscan.org/vert_72/ (accessed on 16 July 2021)), the PDCD4 might be a target of miR-145. The predicted location of binding is in position 590-597 of PDCD4 3ʹ-UTR (Figure 4C). To test this hypothesis, a dual-luciferase reporter assay was performed in HUVECs. Potential binding sites of miRNA-145 in the 3′-UTR of the PDCD4 mRNA were determined by TargetScan. Firefly luciferase activity was significantly reduced when co-transfected with miRNA-145 mimic, indicating that PDCD4 is a target of miRNA-145 (Figure 4C). Remarkably, the miRNA-145 overexpression caused by miRNA-145 mimic transfection down-regulated the PDCD4 mRNA expressions in HUVECs (Figure 4C). Thus, the uptake of EVs by neural stem cell (NSCs) was visualized by labeling EVs with fluorescent lipid dye PKH67. During 12 h incubation of HUVECs with PKH67-labeled EVs, the fluorescent could be well imaged in the cytoplasm location of HUVECs, indicating the HUVECs could efficiently take in EVs derived from VSMCs (Figure 4B). The above results suggested that the LSW could increase the miRNA-145 load in VSMCs-derived EVs, and the LEVs could internalize into HUVECs and target the PDCD4.

### 2.5. LEVs Improved Endothelial Migration by Reducing the oxLDL-Induced PDCD4 Overexpression in HUVECs

The PDCD4 has been defined as a novel programmed cell death factor and is responsible for the malignant proliferation and migration of cells. Hence, the western blot assay indicated that the oxLDL could significantly promote the expression of PDCD4 in the HUVECs (Figure 5A). siRNA was used to knock down the PDCD4 in HUVECs (Figure 5B), and PDCD4 knockdown could inhibit the over-migration of HUVECs with oxLDL-induction (Figure 5C,D). Hence, this result illustrated that the oxLDL might promote migration of HVUECs by up-regulation of PDCD4. The miRNA-145, as a regulator of PDCD4, was loaded into the LEVs. Therefore, we explored the effects of LEVs on the PDCD4 expression in oxLDL-induced HUVECs. In contrast to NEVs, the LEVs could significantly reduce the expression of PDCD4 in oxLDL-induced HUVECs (Figure 5E). Additionally, the qRT-PCR result showed the co-culture with LVEs could decrease the mRNA expression of PDCD4 in the HUVECs, but the not for the NEVs (Figure 5F). In conclusion, the EVs from VSMCs with LSW-treatment could attenuate the oxLDL-induced endothelial pernicious migration.

## 3. Discussion

Previous studies have shown that CVD including hypertension, myocardial infarction, and coronary artery disease are involved in numerous pathophysiological events of dysfunction in the heart and vessels, which has been considered a primary mortality inducement around the world [24]. Atherosclerosis (AS) is a lethal factor to CVD and commonly accompanies an excessive accumulation of lipids in the vascular wall [25]. The endothelium, an important tissue in the vessel, plays a principal role in maintaining the dynamic of vascular tone, angiogenesis, redox reaction, inflammatory, and antithrombotic [26]. Recent studies indicated that endothelial dysfunction, characterized by oxidative stress, chronic inflammation, leukocyte adhesion, and endothelial senescence, should be acknowledged as a vital hallmark of AS [27]. Traditional therapeutic schedule is limited by various side effects, and the edible protein-derived peptides have been explored as a potential nutrient to the cardiovascular system because of its anti-hypertension, ACE inhibition, and vasodilation activities [28]. Our previous study published that the tripeptides LSW from soybean protein hydrolysates could affect the endothelial physiological function through a VSMCs-produced EVs mediating pathway [21]. Further investigation about potential mechanisms of EVs-loading miR-145 on endothelial dysfunction was performed and indicated that the tripeptide LSW could improve oxLDL-induced proliferation, migration, and adhesion by regulating miR-145 packaging in VSMC-derived EVs.

Increasing evidence suggests that the communication between vascular endothelial cells and smooth muscle cells plays a crucial role in the AS, and the extracellular vesicles have served as an indispensable mediator of this interaction [29]. On this basis, a series of natural products and biopeptides also respond to the function of EVs regulation. A recent study indicated that the curcumin could improve the migration and lipid accumulation of VSMCs damaged by LPS-induced endothelial EVs, and the differential expression of miRNAs loading in EVs was detected to explain the potential molecule mechanism [30]. The Paeonol from the radix of Cortex Moutan was proved to improve the inflammatory response of endothelial cells by mediating the miRNA-223 packaging in the monocytes-derived EVs [12]. Liu et al. reported that the polysaccharide of *Dendrobium officinale* could attenuate the inflammatory bowel disease by mediating intestinal EVs with miRNA-433-5p delivering [31]. Consequently, two different EVs from VSMCs with or without peptide LSW incubation were prepared (Figure 2), and the appropriate concentration of LSW was selected not to cause adverse effects on VSMCs and HVUECs (Figure 1). The NTA results showed that the LSW might slightly increase the EVs secretion, but the particle size of LEVs did not be enlarged (Figure 2B). The EV science has found a number of genetic proteins that control EV production, but there is little definitive evidence to link the amount of EV secretion to the phenotypic state of donor cells [32]. The legible mechanism to the pathways of EV secretion will be a long process.

Tripeptide LSW was first explored from glycinin A1bB2-784 as an ACE inhibitor with a low IC_50_ value, which represented a potential anti-hypertensive activity [33]. Subsequently, a Caco-2 transport assay of LSW was established and proved that it was intactly transported across Caco-2 monolayers by the tight junction and peptide transporter 1 (PepT1) mediating paracellular diffusion pathway [34]. Those results might provide a potential that the LSW with a high bioavailability could enter the circulatory system via an oral-absorption pathway. Furthermore, the in vitro vasoactivity of LSW was exhibited by an Ang II-induced VSMCs model, which found that LSW exerted an anti-oxidant and anti-inflammatory activity by reducing the AT1R expression with the Src and ERK1/2 phosphorylation [22]. Hence, the LSW was regarded as a vasoactive peptide. Interestingly, LSW treated alone did not affect oxLDL-induced endothelial proliferation and migration (Figure 3), even though LSW could prompt EVs secretion of VSMCs. However, EVs produced by LSW-induced VSMCs (LEVs) could significantly attenuate oxLDL-induced undesirable migration of VSMCs. Meanwhile, the distinct adhesion ability of LEVs- or NEVs-induced VSMCs has explored a similar result due to a moderate expression of adhesion protein (Figure 3D,F). Although the tripeptide LSW showed some intervening effects on the physiological regulation of EVs in this study, more investigation need to be performed, such as the relationship between biopeptides and EVs-secretion pathways, or the mechanisms of biopeptides intervention miRNA intracellular synthesis, etc.

The miRNA-145 has been verified to load into EVs and decrease the atherosclerotic lesion formation in KLF2-expressing VECs [35]. However, the communication between VECs and VSMCs is not monodirectional or simple from VECs to VSMCs, and some changes occurred in VSMCs might interfere the VECs as a downstream form of the conversation in turn [29]. In the present study, we found that LSW not only increased the expression of miRNA-145 in donor VSMCs but also elevated the loading of miRNA-145 in the VSMCs-derived EVs (Figure 4A). Remarkably, although LSW treatment altered the miRNA-145 expression, it did not induce special responses of VSMCs in proliferation and migration (Figure 1). The internalization of EVs into cells was a precondition to deliver miRNAs to recipient cells. The fluorescent staining assay displayed that EVs were traced into HUVECs with the extension of time (Figure 4B). The programmed cell death protein 4 (PDCD4) has been defined as the regulator of myriad cellular events, and a recent study indicated that PDCD4 played a crucial role in the formation of coronary AS plaque and promoted the production of IL-6 and IL-8 in the VSMCs [36]. The miRNA-16 was explored as a target of PDCD4 and participated in suppressing the activation of inflammatory macrophages by the MAPK and NF-κB signaling pathway in AS [37]. However, it is worth exploring whether the antagonistic effect of LEVs on oxLDL-induced endothelial dysfunction stems from PDCD4. Luciferase assay was used to test the interaction between miRNA-145 and PDCD4 in the HUVECs (Figure 4C). The oxLDL elevated the expression of PDCD4 in HUVECs, and PDCD4 knockdown could obviously reduce the migration of oxLDL-induced HUVECs (Figure 5). Those results supported that PDCD4 mediates the damage effects of oxLDL, which is consistent with other studies. Bai et al. suggested that the direct involvement of PDCD4 in oxLDL-induced stress granules through its RNA-binding activity [38]. Another study reported that the miRNA-21 directly targeted the PDCD4 of HUVECs and controlled apoptotic proteins expression [39]. The negative regulatory relationship between miRNA-145 and PDCD4 was further verified in the EVs co-incubation study (Figure 5E,F). Distinct from NEVs, LEVs significantly down-regulated the PDCD4 expression and simultaneously occurred at the level of transcription. Although it may be prudent to suggest that the regulation of EVs on HUVECs is via a miRNA-mRNA sponge, the uniqueness of this relationship need some further work to make it clear.

## 4. Materials and Methods

### 4.1. Chemicals

DMEM cell medium, phosphate-buffered saline (pH 7.4), RPMI-1640 media, and fetal bovine serum (FBS) were obtained from Gibco (Carlsbad, CA, USA). In addition, 0.25% Trypsin-EDTA and penicillin-streptomycin solution were purchased from Sigma-Aldrich (St. Louis, MO, USA). Serum-free media (for exosome culture) was purchased from Umibio (Shanghai, China) Co. Ltd. oxLDL was purchased from Solarbio Life Sciences (Beijing, China). The RIPA buffer, cck-8 kit, DAPI dye, and dihydroethidium (DHE) were purchased from the Beyotime Institute of Biotechnology (Shanghai, China). A protein quantification kit with bicinchoninic acid (BCA) protein was purchased from ThermoFisher Scientific (Waltham, MA, USA). The tripeptides Leu-Ser-Trp (LSW) were purchased from GL Biochem (Shanghai) Ltd. (Shanghai, China). The purity of the peptide was determined by HPLC (99.9% for LSW) according to the manufacturer. After dissolving in 1 × PBS, peptides were aliquoted and stored at −20 °C for cell culture experiments.

### 4.2. Cell Culture

Referring to our previous study [21], the VSMCs and HUVECs cell line were a kind gift of Professor Zedong Jiang (Jimei Universtiy, Xiamen, China) and were used in this study between passages 5 and 15. The cells received were maintained in DMEM with 10% FBS containing 100 U/mL penicillin, and 100 g/mL streptomycin at 5% CO_2_ and 37 °C. The U937 monocytes, a human leukemic monocyte lymphoma cell, were purchased from Nation Collection of Authenticated Cell Cultures (Shanghai, China), and cultured in RPMI-1640 media with 10% FBS. For induction experiments, HUVECs were stimulated with 100 μg/mL oxLDL with or without LSW/EVs (1 μg protein/mL) pre-incubation. The cell proliferation was assessed by a CCK-8 kit following the instructions. Briefly, cells were cultured in 96-well plates with 80% confluency, and then co-incubated with various concentrations (0.1–100 μmol/L) of LSW for another 24 h. The CCK-8 reagent was added to plates at 100 μL/well, and the plate was kept at 37 °C for 2 h. Then, the optical density at 562 nm wavelength was tested by a VARIOSKAN FLASH microplate reader (Thermo Fisher Scientific, Waltham, MA, USA).

### 4.3. Dihydroethidium Staining and Wound Healing Assay

After stimulation, the HUVECs with or without LSW/EVs pre-incubation was loaded with 1 μmol/L DHE dye and maintained for 15 min at 37 °C. After that, the medium containing dye was removed and the cells were gently washed with PBS. The intracellular production of ROS was observed using a fluorescent microscope. The wound-healing assay was used to assess the cell migration ability. The cells were inoculated in a 6-well plate with LSW or EVs pre-incubation, and then a scratch was made on a uniform confluent layer using a sterile micropipette tip. The photographs of the scratches were taken immediately at 0 and 24 h, respectively.

### 4.4. Adhesion of U937 Monocytes to HUVECs

The adhesion assay was performed by a co-culture with U937 monocytes and HUVECs as previously showed [40]. Briefly, the U937 monocytes (2 × 10^4^ cells/mL) were labeled with 5 μmol/L calcein-AM for 60 min at 37 °C. Then, the U937 cells were centrifuged to abandon extra stain and added to culture media containing LSW- or EVs-pretreated HUVECs with or without oxLDL induction for 2 h. After that, the free U937 cells were removed by washing the plates two times. Fluorescence photographs were visualized through a fluorescence microscope.

### 4.5. Isolation and Characterization of Extracellular Vesicles

The EVs from the cells culture medium were isolated by an ultra-centrifuge process according to a method described by Théry et al. with small modifications [41]. The VSMCs were incubated with or without 50 μmol/L LSW for 24 h, and the cells were washed three times with PBS. Afterwards, the fresh media for exosome culture (serum-free) were added and continued to culture for 48 h. Then, the culture medium was collected and centrifuged at 300× *g* for 15 min and then at 3000× *g* for 30 min. The supernatants were harvested to extract the EVs. The supernatants containing extracellular vesicles were centrifuged at 10,000× *g* for 60 min at 4 °C to remove cells and debris, and thereafter at 100,000× *g* for 60 min. The EV particles were concentrated at the bottom of tubes and resuspended with 100 μL PBS. The EVs from normal VSMCs without LSW incubation are named NEVs, and the EVs from LSW-induced VSMCs are named LEVs. The EVs were stored at −80 °C for further research. The EVs were quantified through a BCA kit.

The morphology of EVs was observed by TEM testing as described previously [41]. Briefly, the EVs suspension was mixed with an equal volume of 4% paraformaldehyde and deposited on Formvarcarbon-coated EM grids. The photograph was imaged with a transmission electron microscope (Hitachi, Tokyo, Japan). The sizes of EVs were analyzed through a nanoparticle tracking analysis by NanoSight NS3000 instrument (Malvern Panalytical Ltd., Worcestershire, UK) as previously described [42]. The biomarker protein CD9, CD81, and Tsg101 were identified by western blot assay.

### 4.6. The EVs Labeling and Tracer Observations

To further observe whether the EVs were taken up by HUVECs or not, the EVs were labeled with a green fluorescent dye PKH67, and then co-cultured with HUVECs. In brief, the EVs were resuspended in PBS with 100 μM PKH67 dye for 1 h at 37 °C, and then were co-cultured with HUVECs in a 6-well plate at 37 °C for 6 h. After incubation, the HUVECs were gently washed by PBS and fixed with 4% paraformaldehyde for 15 min at room temperature. The 0.1% Triton X-100xw (Beyotime Institute of Biotechnology, Shanghai, China) was used to permeabilize the HVUECs, and then the cells were stained with DAPI solution for 5 min. The fluorescence images were obtained and used to observe the trace of EVs at different time points.

### 4.7. Cell Transfection

The cells were transfected with synthesized siPDCD4 packaging in Lipofectamine 3000 (Invitrogen, Carlsbad, CA, USA) as previously published [43]. Briefly, the HUVECs were plated onto 6-well plates at a density of 5 × 10^4^ cells/well and allowed to grow overnight. The siPDCD4 or empty vector with 5 μL lipofectamine was co-incubated with cells. After transfection, cells were cultured at 37 °C for 24 h to 72 h, then the cells were harvested and lysed to extract protein or RNAs.

### 4.8. Western Blot and qRT-PCR

The HUVECs were lysed with RIPA buffer containing proteinase inhibitors. Total proteins were quantified using the BCA assay kit, and the equal protein quantity of lysates was separated by SDS-PAGE electrophoresis and transferred to PVDF membranes for analysis. The membranes were blocked with 6% BSA and then probed overnight with primary antibodies and HRP-conjugated anti-IgG at 4 °C. The enhanced chemiluminescence kit (Beyotime Institute of Biotechnology, Shanghai, China) was used to test the blots. Western blot bands were scanned and saved.

The Invitrogen TRIzol reagent was used to extract RNAs from cells according to the manufacturer’s instructions. The cDNA was synthesized by a Transcriptor First-Strand cDNA Synthesis System (Applied Biosystems, Branchburg, NJ, USA). The primer sequences used for the real-time PCR analysis were as follows: miRNA-145 forward: 5′-ACGGTCCAGTTTTCCCAGGAATCCCT-3′; PDCD4 forward: 5′-AGGTTGCTAGATAGGCGGTC-3′; U6 forward: 5′-CTCGCTTCGGCAGCACA-3′; GAPDH forward: 5′-AACGACCCCTTCATTGACCTC-3′ [44]. For miRNA detection, a universal reverse primer from Invitrogen was used. PCR amplification was carried out on a C1000 Touch Fast Real-Time PCR system (Bio-Rad Laboratories, Inc., Hercules, CA, USA) with the primers purchased from Guangzhou RiboBio Company (Guangzhou RiboBio CO. Ltd., Guangdong, China).

### 4.9. Luciferase Activity Analysis

For identification of the binding site between miR-145-3p and PDCD4, cells were transfected with a luciferase construct containing PDCD4 with the wild-type or a mutated version of the binding site, co-transfected with miR-145-3p mimic or negative vector. Before transfection, the cells were seeded in a 96-well plate at a density of 1 × 10^4^ cells/well. Then, medium of each well was replaced with fresh liquid. After that, the transfection mixture was prepared and suppled into the plate according to Lipofectamine 3000 instructions. The luciferase activities were measured using a Dual-Luciferase kit (Promega, Madison, WI, USA) according to the manufacturer’s instructions after 48 h of transfection at 37 °C.

### 4.10. Statistical Analysis

Statistical analyses were performed for all results by the SPSS 21.0 software package. All results data were presented as the mean ± SD unless otherwise specified. Differences between means were evaluated by one-way analysis of variance (ANOVA) for multiple comparison tests. *p* < 0.05 was defined as statistically significant.

## 5. Conclusions

In conclusion, this study provided evidence that the antihypertensive peptide LSW protected oxLDL-stimulated endothelial proliferation and migration via VSMCs-derived EVs enriched with miRNA-145. LSW could increase the miRNA-145 expression in VSMCs and loading in VSMCs-produced EVs. The EVs from LSW-stimulated VSMCs attenuated oxLDL-induced endothelial dysfunction by targeting the PDCD4 expression in HVUECs. Based on these results, the peptide LSW is considered a potential agent for mitigating lipid accumulation-induced vascular endothelial dysfunction, and the EVs-mediated transfer of miRNA-145 from VSMCs to VECs may be an effective therapeutic target for endothelial dysfunction.

## Figures and Tables

**Figure 1 molecules-27-07025-f001:**
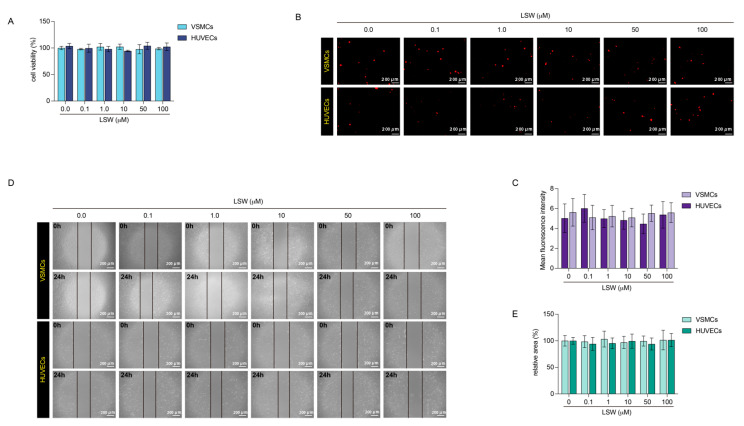
The effects of tripeptide LSW on proliferation and oxidative stress of VSMCs and HVUECs. The VSMCs and HUVECs were treated with LSW. (**A**) The CCK-8 assay was used to test the proliferation VSMCs and HUVECs with or without LSW incubation. (**B**) Images of dihydroethidium staining in the VSMCs and HUVECs with or without LSW incubation in different concentrations (100× magnification, scale bar: 200 μm). (**C**) The mean fluorescence intensity of the DHE-staining images was analyzed by Image J software. (**D**) The migration of VSMCs and HUVECs was evaluated by a wound-healing assay. The scratch of the wound healing was recorded at 0 and 24 h (100× magnification, scale bar: 200 μm). (**E**) The relative area of the wound healing scratch at 24 h was measured by ImageJ software (version 2.1.0) and the results of each treatment group were shown to the corresponding control (without LSW treatment). Data represent the mean ± SD of three independent experiments.

**Figure 2 molecules-27-07025-f002:**
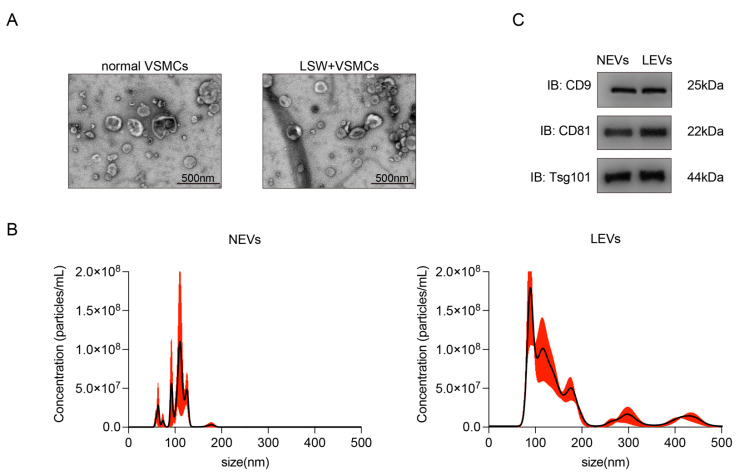
Characterization of extracellular vesicles produced from VSMCs with or without LSW treatment. (**A**) TEM analysis of EVs from normal VSMCs without LSW treatment (NEVs, left panel) and LSW-treated VSMCs (LEVs, right panel). Scale bar, 500 nm. (**B**) Particle sizes of NEVs and LEVs, as measured with the NanoSight analysis. (**C**) The expression of EV-specific markers CD9, CD81, and Tsg101 was measured with western blot analysis.

**Figure 3 molecules-27-07025-f003:**
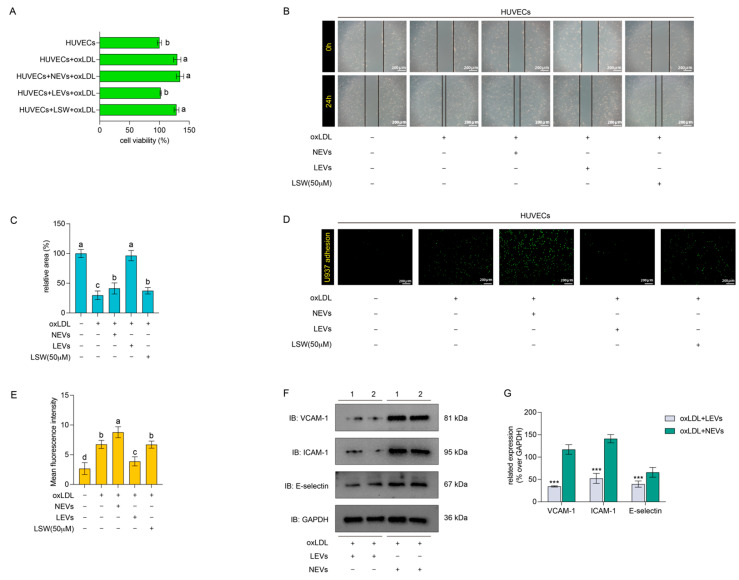
The different regulations of small EVs and LSW on the proliferation, migration, and adhesion of HUVECs. The HUVECs were pre-treated by EVs or LSW for 12 h prior to stimulation with oxLDL. (**A**) The relative cell viabilities of the HUVECs pre-treated with EVs or LSW in presence of oxLDL, and it is significantly different (*p* < 0.05) with different alphabetical mark. (**B**) The macroscopic images of wound healing on 0 and 24 h (100× magnification, scale bar: 200 μm). The scratch was created on the HUVECs and the wound closure was monitored with the application of NEVs, LEVs, and LSW. (**C**) The relative area of the wound healing scratch at 24 h was measured by ImageJ software and the results of each treatment group were shown to the normal HUVECs without special treatment, and it is significantly different (*p* < 0.05) with different alphabetical mark. (**D**) The fluorescence images of EVs and LSW on U937 cell adhesion to the oxLDL-induced HUVECs (100× magnification, scale bar: 200 μm). HUVECs were incubated with NEVs, LEVs, and LSW for 12 h before stimulating with oxLDL (100 μg/mL). (**E**) The mean fluorescence intensity of the adhesion images was calculated by Image J software, and it is significantly different (*p* < 0.05) with different alphabetical mark. (**F**) The adhesion molecules VCAM-1, ICAM-1, and E-selectin were detected by western blot in the oxLDL-induced HUVECs with EVs and LSW treatment. (**G**) The relative expression of proteins were calculated by densitometry and normalized to their loading controls, *** *p* < 0.05.

**Figure 4 molecules-27-07025-f004:**
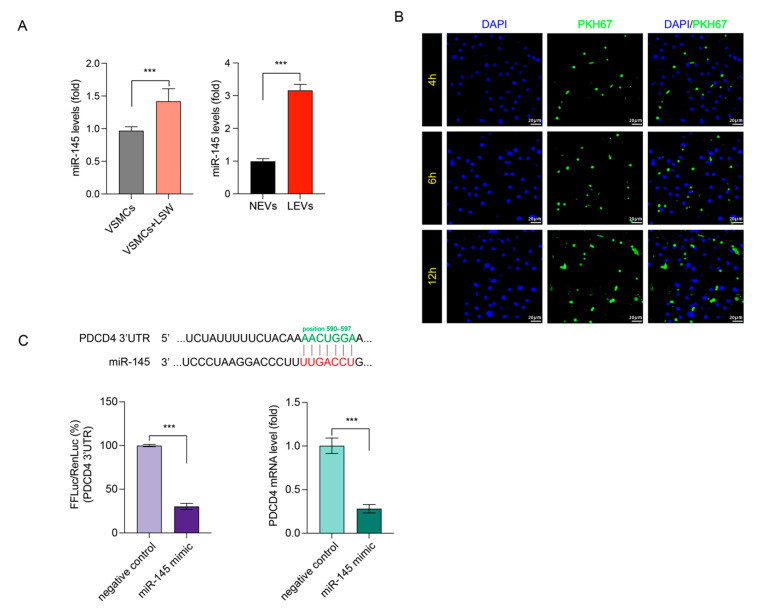
Tripeptide LSW induced miRNA-145 to package in EVs and target the PDCD4 expression in the HUVECs by internalizing into cells. (**A**) The qRT-PCR was used to detect the relative expression of miRNA-145 in the VSMCs and VSMCs-derived EVs with or without LSW induction, *** *p* < 0.05. (**B**) The microscopic images showed uptake of the EVs from LSW-induced VSMCs by the HUVECs in a time-dependent manner (200× magnification, scale bar: 20 μm). The cell nuclei were stained by DAPI in blue and the EVs were stained by PKH67 in green. (**C**) The binding position between miRNA-145 and PDCD4 was achieved from TargetScan database. The HUVECs were transfected in combination with miR-155 mimic or negative control miRNA, *** *p* < 0.05. Luciferase activity was determined using a dual-luciferase reporter assay kit. The mRNA relative expression of PDCD4 in the HUVECs with miR-155 mimic or negative control miRNA incubation was quantitated. Results were presented as mean ± SD, *n* = 5 for each group.

**Figure 5 molecules-27-07025-f005:**
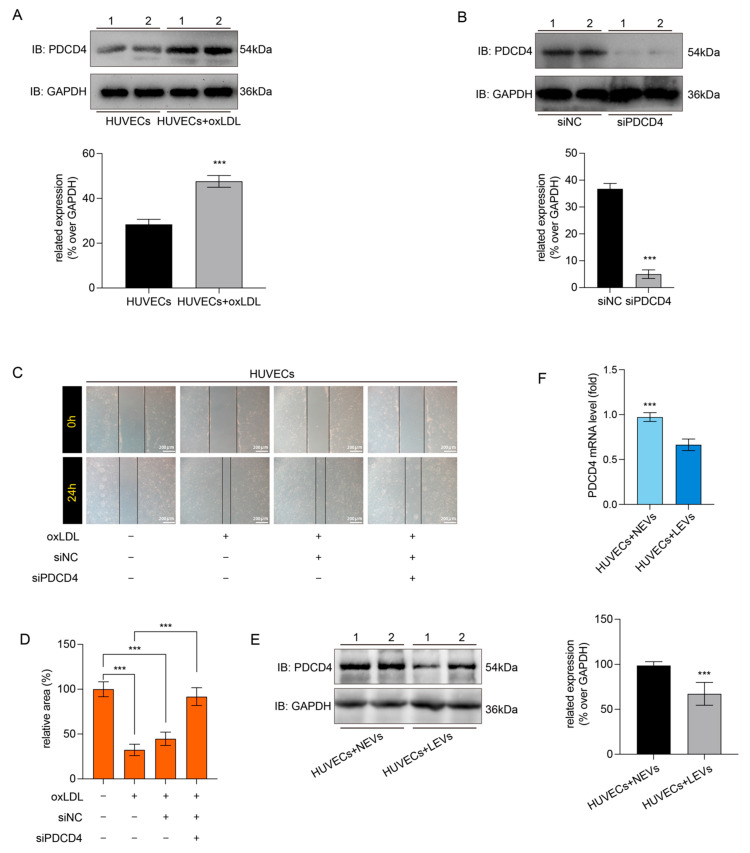
The EVs from LSW-induced VSMCs ameliorate the malignant migration of oxLDL-induced HUVECs by mediating the PDCD4 down-regulation. (**A**) The oxLDL increased the PDCD4 expression by a western blot analysis. The GAPDH was used as the loading control. The immunoblotting images were represented by densitometry as repetitive bands, *** *p* < 0.05. (**B**) The PDCD4 expression in the HUVECs with siRNA-PDCD4 (siPDCD4) or siRNA-negative control (siNC) was detected. The immunoblotting images were represented by densitometry as repetitive bands, *** *p* < 0.05. (**C**) The PDCD4 knockout reduced the malignant migration of oxLDL-induced HUVCEs via a wound-healing assay (100× magnification, scale bar: 200 μm). (**D**) The relative area of the wound healing scratch at 24 h was measured by ImageJ software, and the results of each treatment group were shown to the normal HUVECs without special treatment, *** *p* < 0.05. (**E**) The expression of PDCD4 in the HUVECs with NEVs or LEVs incubation. The immunoblotting images were represented by densitometry as repetitive bands, *** *p* < 0.05. (**F**) The mRNA expression of PDCD4 in the HUVECs with NEVs or LEVs incubation, *** *p* < 0.05. Results were presented as mean ± SD, *n* = 5 for each group.

## Data Availability

Not applicable.

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
