# Peer review of "Tripeptide Leu-Ser-Trp Regulates the Vascular Endothelial Cells Phenotype Switching by Mediating the Vascular Smooth Muscle Cells-Derived Small Extracellular Vesicles Packaging of miR-145"

_molecules, 2022, doi:10.3390/molecules27207025_

Round 1
Reviewer 1 Report
In general this is a very good paper. The hypothesis that the tripeptide LSW can reduce the inflammatory response of endothelial cells via microvesicles released from smooth muscle cells is a very interesting one.
The paper is not the whole story, nor does it completely prove the hypothesis. The paper opens up a whole line of enquiry and generates more questions than it answers. That, is a sign of a good paper.
The methodology looks reasonably sound and the experimental evidence would support the sub-hypothesis that LSW-treated smooth muscle cells produce extracellular vesicles containing miR-145 that are effective in reducing the effects of oxidised LDL on HUVEC migration.
The Discussion could use some more work on the English language expression and a reduction in the enthusiastic claims for efficacy of natural products on a general basis, as the evidence does not support that generalisation. The sentence on page 18 lines 350 to 352 of the proof: "This is not one exception that natural product generates the EVs as a different function than normal cell production, which is vitally responsible of miRNAs for this phenomenon." is especially dangerous and should be removed or rewritten to satisfy what has been shown by the data in this paper.
In page 18 line 360 the authors refer to the evidence showing a lack of "malignant responses" of the smooth muscle cells. Malignancy was not investigated and this description should be removed. No increase in cell proliferation was noted. Cell proliferation is, of itself, not a sign of malignancy.
The concluding sentence of the Discussion ("Based on these results, the peptide LSW is considered a potential agent for mitigating lipid accumulation-induced atherosclerosis, and EV-mediated transfer of miRNA-145 from VSMCs to VECs may be an effective therapeutic target for endothelial dysfunction in atherosclerosis") is also an overstatement as the development of atherosclerosis and endothelial cell 'dysfunction' may be related, but the former is not proven to be caused by the latter.
Reviewer 2 Report
In this manuscript, Tianyuan Song et al describe in a very adequate way the
Tripeptide LSW has a protective effect against endothelial dysfunction on vascular endothelial cells via vascular smooth muscle cells-derived miRNA-145 19 packaged in EVs.
All the evidence shown here is very interesting and relevant, however, they are some minor points corrections:
1.- In Western blort assays, please perform densitometry.
2.- In line 284 it says "western" instead of "Western".
3.- In line 291 it is not very appropriate to use the adverb, obviously
4.- In figure 1b, add a description to the image
